# Foam Formation and Interaction with Porous Media

**Phillip Johnson [1], Mauro Vaccaro [2], Victor Starov [1] and Anna Trybala [1,\*]** 

1   Department of Chemical Engineering, Loughborough University, Loughborough LE11 3TU, UK;
    P.Johnson@lboro.ac.uk (P.J.); V.M.Starov@lboro.ac.uk (V.S.)
2   Procter & Gamble, Temselaan 55, B-1853 Grimbergen, Brussels, Belgium; vaccaro.m@pg.com
\*   Correspondence: A.Trybala@lboro.ac.uk

**Abstract:** Foams are a common occurrence in many industries and many of these applications require the foam to interact with porous materials. For the first time interaction of foams with porous media has been investigated both experimentally and theoretically by O. Arjmandi-Tash et al. It was found that there are three different regimes of the drainage process for foams in contact with porous media: rapid, intermediate and slow imbibition. Foam formation using soft porous media has only been investigated recently, the foam was made using a compression device with soft porous media containing surfactant solution. During the investigation, it was found that the maximum amount of foam is produced when the concentration of the foaming agent (dishwashing surfactant) is in the range of 60–80% m/m. The amount of foam produced was independent of the pore size of the media in the investigated range of pore sizes. This study is expanded using sodium dodecyl sulphate (SDS), which has the same critical micelle concentration as the commercial dishwashing surfactant, where the foam is formed using the same porous media and compression device. During the investigation, it was found that 10 times the critical micelle concentration (CMC) is the optimum concentration for a pure SDS surfactant solution to create foam. Any further increase in concentration after that point resulted in no further mass of foam being generated.

**Keywords:** foams; porous media; wetting; surfactant; foam formation; SDS

## 1. Introduction

Liquid foams are used in many industries and a majority involve foam interaction with porous materials, for example, firefighting, enhanced oil recovery, house and dish cleaning products [1–5]. The way foams interact with a porous material only recently was investigated and is still to be completely understood [6]. Recently a focus has been put on understanding the interaction of dishwashing solution with soft porous media [7,8] (sponge) using the home made compression device [8]. This specific new home-made devise allows investigating how properties of the soft porous media and the surfactant solution effect the foam formation.

An important industrial application of foams is enhanced oil recovery [4,9–11] where foams can be used to increase the amount of crude oil that can be extracted from an oil field. In this process usually, gas flooding is used where $CO_2$ and/or $N_2$ are injected into the reservoir [10,11]. The issue that arises is that these gases are normally less viscous, less dense and highly mobile than both crude and water resulting in gas channeling [10,11]. This is when the injected gas goes through high permeability layers and fractures in the porous structure inducing poor displacement of the crude oil. Another issue is gravity override, this is a phenomenon of multiphase flow within a reservoir where less dense fluid flows on top of the denser crude further reducing displacement [12]. This means to increase the displacement the gas channeling and gravity override has to be reduced; this is done by creating a foam to work as a mobility control [10–12]. The foam has an apparent viscosity greater than the

injected gases hence lowering the mobility allowing the solution to enter areas previously untouched when using gas alone leading a greater amount of oil recovered [12].

Liquid foams are formed when a gas is injected into a liquid and the rate of bubble generation is faster than the drainage of the liquid from the between the bubbles [13–17]. For this to occur the liquid-air bubble interfacial tension must be reduced, which is done by adding a surface-active agent like surfactant and/or polymers. Surfactants are used to reduce the surface tension by adsorbing at the air/liquid interface [18], slowing down coalescence and increase the surface viscosity at the interface. The presence of surfactants results also in an increased interfacial viscosity, which helps provide additional resistance to film thinning between bubbles and rupturing and strengthens the foam [13]. Below sodium dodecyl sulphate (SDS) is used, an ionic surfactant, which forms charged layers to form an additional repulsive force between bubbles [19,20].

Critical Micelle Concentration (CMC) is the surfactant concentration at which micelles start to form. A micelle is a collection of the surfactant molecules where the hydrophobic tails associate together to form the inside of a micelle, whilst the hydrophilic heads all form the circumference of the micelle [13,20,21]. All liquid foam generation methods require the same principle of generating air bubbles within a liquid. Each method differs by how the energy is provided to produce the foam. A general classification of the foaming methods is presented below [14]:

- Mechanical action (Breaking up bubbles under shear, bubbling through an orifice)
- Phase Transition (Supersaturation of the liquid in aerosol cans, champagne bottles)
- Electrochemical (Electrolysis)
- Chemical Reaction (Carbonated drinks)
- Biological (Yeast)

The investigated below method of foam formation using compression of soft porous materials is classified as mechanical action, as the foam is formed by the pressure reduction when compressing the soft porous material (sponge). Currently little is known about the mechanism behind this method in terms of where the foam forms or the factors that influence foam formation.

Only recently work has been focused on how a foam interacts with a porous substrate. The first model of foam drainage in contact with porous media was introduced in [22,23]. It was found that there are three different regimes of the process: rapid, intermediate and slow imbibition depending on the relation of rates of drainage and imbibition into the porous substrate (Figure 1). In the case of the slow regime of imbibition, the liquid layer forms temporary at the foam-porous substrate interface (Figure 1c).

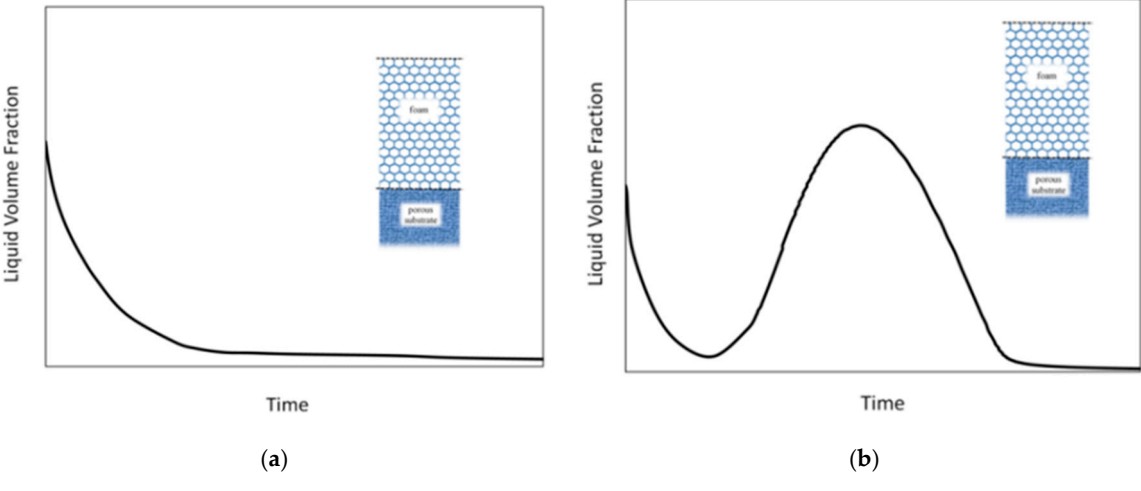

**Figure 1.** *Cont.*

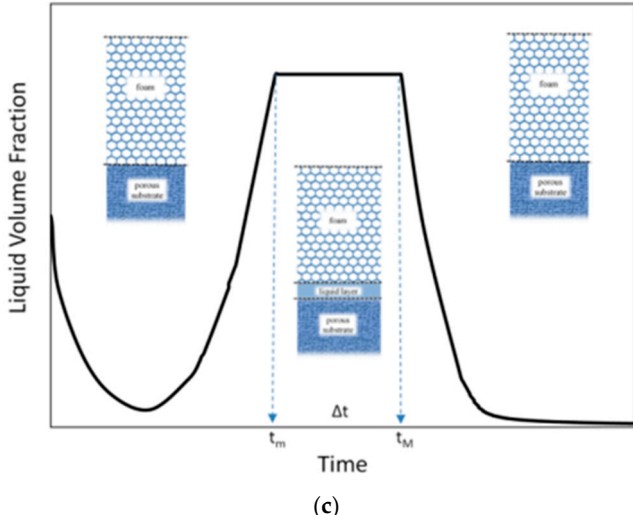

(c)

**Figure 1.** Schematic of the time evolution of liquid volume fraction at foam/porous substrate interface **c** in three different regimes of (**a**) rapid imbibition into the porous substrate: the liquid volume fraction gradually deceases; (**b**) intermediate imbibition into the porous substrate: the liquid volume fraction goes via max but this max is below critical when liquid layer starts forming; (**c**) slow imbibition into the porous substrate: at some moment $t_m$ the free liquid layer starts forming at the foam/porous substrate interface and this layer disappeared at the moment $t_M$.

An extension of this theory and experimental verification was undertaken in [24] were foam drainage placed on thin porous substrates was investigated experimentally and theoretically verifying the three regimes discussed in [22]. This work leads to using porous media as a method to force drainage in a micro gravity environment [25] where the gravitational drainage is impossible and the drainage is forced by the capillary suction into the porous media. In this investigation, the model was verified by experimental investigations. The new method for drying foams in microgravity conditions was suggested [24].

Whereas unlike previous investigations that have focused on the interaction of foam with rigid porous media the results presented below concentrate on foam production with soft porous media. The foam is produced using a new homemade device designed to simulate compression of the soft porous media, compression of sponges in household applications (dishwashing, bathing etc.). The way the solutions interact with the porous media is an important property when considering how much of the surfactant is available for foam production. Finally, new results using SDS solutions are introduced and support observations made with the commercial product, the issues caused by the complexity of the commercial dishwashing solution are indicated.

## 2. Materials and Methods

Below the results on foam formation using commercial surfactant solutions are presented [3]. For this purpose, a compression device was designed (Figure 2). The device consists of two parallel plates controlled by pistons which can move to bring the plates together (Figure 2). The pistons are regulated by opening and closing valves governing the air flow through the system, each plate is controlled by a piston and the pressure of each can be changed using the connected computer. The pressure rate applied to the sponge can be controlled by a regulator to vary the air flowing to the piston which determines the velocity of the plates. During the study, the air flow was kept constant, and the pressure of the pistons was set to 2.62 bar (2672.66 N).

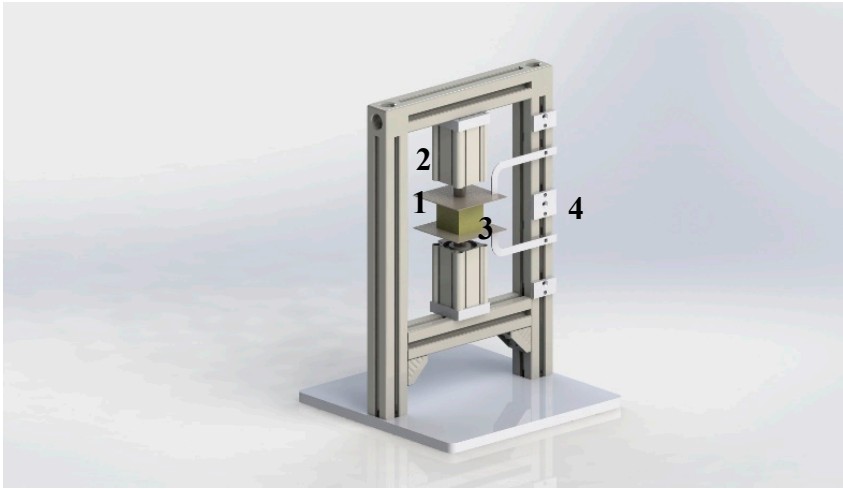

**Figure 2.** Experimental compression device. 1—Metal plates; 2—Pistons 3—Porous material; 4—Controls varying the pressure to each piston [3].

Three sponges were used as a model of soft porous material and are discussed below. Commercially available dishwasher, car and audio sponges were used. The sponges are stated to be 100% polyester by the manufacturer. A scanning electron microscope (SEM), Hitachi Benchtop SEM, was used to evaluate the properties of each sponge. The proprietary software MatLab (R2019b) was utilized to obtain each sponges porosity. Comparing the total area of the image to the area of the dark contrast pores where the walls of the pores are light contrast allows the porosity to be calculated Using up to 45 pores within each sample the ImageJ software enabled the average pore size to be determined. The SEM images of all of the sponges are presented in Figure 3.

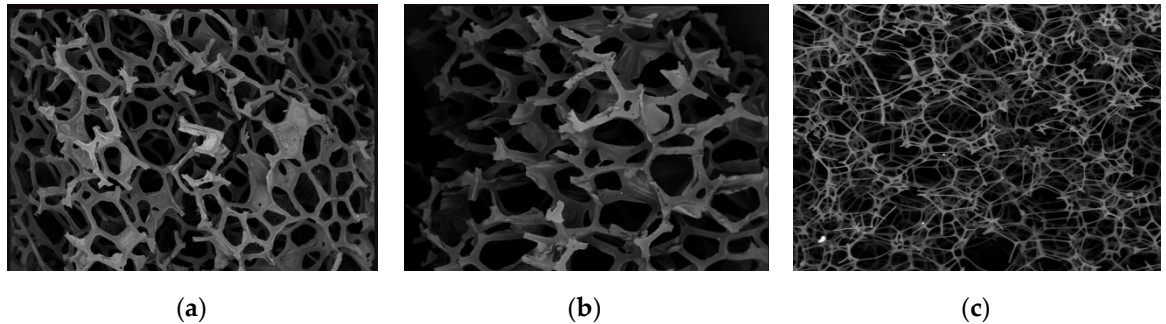

(**a**)      (**b**)      (**c**)

**Figure 3.** SEM images of (**a**) car sponge; (**b**) dishwasher sponge; (**c**) audio sponge [3].

The values of pore size and porosity of each of the sponges are presented in Table 1

**Table 1.** The pore size and porosity of the sponges [8].

| Sponge Type | Pore Size (mm) | Porosity |
|---|---|---|
| Dishwashing sponge | 0.2953 ± 0.0704 | 0.694 ± 0.013 |
| Car sponge | 0.3017 ± 0.0723 | 0.690 ± 0.006 |
| Audio sponge | 0.0928 ± 0.0281 | 0.692 ± 0.005 |

The total mass of the surfactant solution for each concentration was 30 g, which is the total mass of liquid that the sponge can hold when saturated without leakage. After adding 30 g of surfactant solution the sponge was compressed. The compression was held for 5 s and then the pressure was

released. The compressions were repeated 5 consecutive times; the foam obtained after this process was collected and weighted.

To expand on the work previously conducted with commercial surfactant a pure surfactant is used with the same sponges. The surfactant used is Sodium Dodecyl Sulphate (SDS), from Fischer Scientific UK, which is an anionic surfactant where the critical micelle concentration (CMC) is 8.2 mMol/L or 0.24% (by mass) when dissolved in water. Concentrations investigated ranged from 0.5 CMC to 50 CMC.

The pore size and porosity of the sponges were previously measured above and in [8] using an SEM device. The values for the sponges investigated shown in Table 1 (see also Figure 3).

The sponge compression device which was previously used in [8] (Figure 2). The total mass of the SDS solution for each concentration was 30 g the same mass used for the commercial dishwashing solution.

Hard water is a mixture of distilled water with salts that could be found in tap water, the measurement of hardness is in German degrees (dH) where 1 dH is 0.05603 parts per million. As well as sponge type and amount of surfactant used the amount of salt (hardness) of water was investigated, the hard water produce was 15 dH hard water. Allowing the effect salt content has on the foamability. The salts added and the amounts in grams added are displayed by Table 2.

**Table 2.** The salts and amount in grams used to make 2 litres 15dH hard water.

| Salt | Mass (g) |
|---|---|
| $CaCl_2 \cdot 2H_2O$ | 0.56 |
| $MgCl_2 \cdot 6H_2O$ | 0.30 |
| $NaHCO_3$ | 0.28 |

## 3. Results and Discussion

### 3.1. Foamability of Soft Porous Media using Compression

The mass of foam produced compressing the sponges d was monitored for the commercial surfactant concentration in the range $10 - 100\%$ m/m. A mathematical fitting was performed to describe the trend observed for the mass of foam produced with each of the sponge types. Starting with the fact that initially $M\infty(c)$ dependency increases with surfactant concentration, that is, $M\infty(c) \sim \lambda c$, where $\lambda$ is a constant. However, if the concentration increases the slope of $M\infty(c)$ dependency eventually decreases and goes through the max value (Figure 4). The total amount of water plus surfactant remains constant according to our experimental procedure. As the amount of surfactant increases the amount of water decreases and so does the foamability, that is, $M\infty(c)$. However, even at concentration of surfactant equal to 1 (pure surfactant) the foamability does not go to zero. Indicating that the commercial surfactant used still contains some amount of water, which determines the finite value of foamability even at 100% concentration of surfactant) "pure surfactant". The real water concentration is not 1-c but $\beta$-c, where $\beta > 1$ and includes both the "real" water concentration and unknown concentration of water inside the "pure surfactant".

The latter consideration shows that the $M\infty(c)$ dependency can be presented as (Equation (1))

$$M_\infty(c) = \lambda c(\beta - c) \tag{1}$$

At c = 1 using Equation (4) it was found that $M_\infty(1) = \lambda(\beta - 1) = M_{min}$ or $\lambda = M_{min}/(\beta - 1)$. At the concentration $c = c_{max}$ the dependency $M\infty(c)$ reaches the maximum value when the derivative of

this dependency is equal to zero. This allows the β value to be determined: β = cmax/2. Hence, the dependency, $M_\infty(c)$ can be written as (Equation (2)):

$$M_\infty(c) = \frac{M_{min}}{(2c_m - 1)}(c(2c_m - c)) \tag{2}$$

where $M_\infty(c)$ is the total amount of foam generated, $c_m$ is the concentration of when $M_\infty(c)$ is at its maximum value, c is the concentration of surfactant and $M_{min}$ is the foam mass when 100% surfactant is used. In Figure 4 the schematic presentation of dependency of $M_\infty$, the total surfactant concentration during the sponge compression is demonstrated.

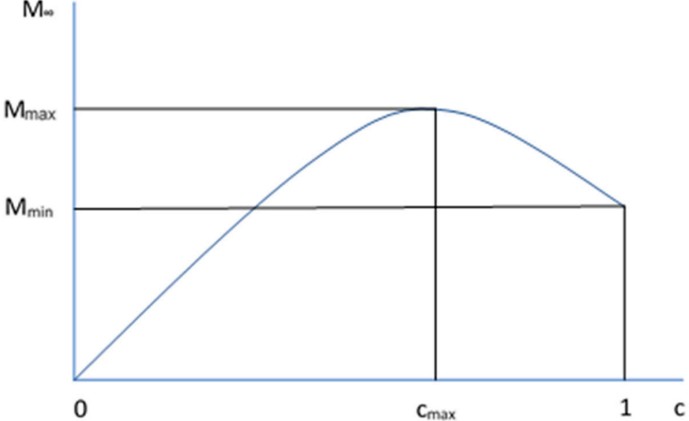

**Figure 4.** Schematic representation of the $M_\infty$ dependency for the total surfactant concentration during the sponge compression [8].

Using Equation (2) a fitting is plotted alongside the experimental results gathered for the dishwasher sponge (Figure 5). Very good agreement between experimental results and a mathematical model was found.

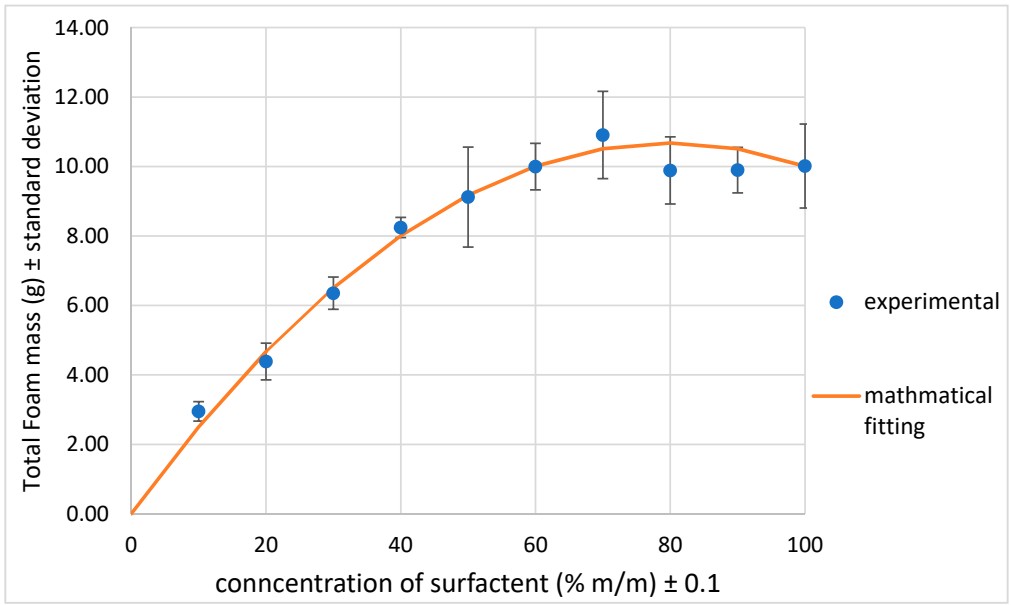

**Figure 5.** Total amount of foam generated for each concentration of surfactant fitted according to Equation (2) with $M_{min}$ = 10.01, $c_{max}$ = 0.8 [8].

Figure 6 shows the foam mass produced using the compression device for the car and dishwasher sponges with distilled and 15 dH hard water solutions.

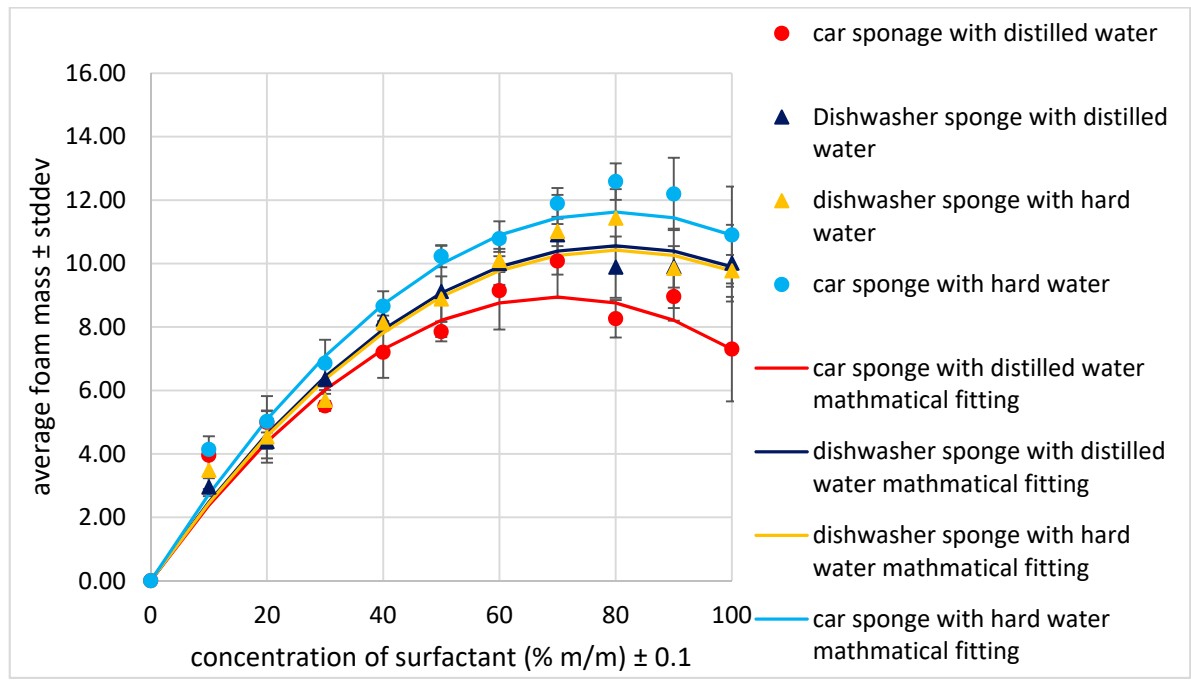

**Figure 6.** Average total foam mass for both the Dishwasher sponge and the car sponge for each concentration of surfactant. Fitted according to Equation (2), with $M_{min}$ = 7.30 and $c_{max}$ = 0.7 for distilled water with dishwasher sponge, $M_{min}$ = 9.77 and $c_{max}$ =0.8 for hard water with car sponge, $M_{min}$ = 10.01 and $c_{max}$ = 0.7 for distilled water with dishwasher sponge and $M_{min}$ = 10.90 and $c_{max}$ = 0.8 for hard water with car sponge [8].

Figure 6 shows that the amount of foam produced increases until 60–80% m/m concentration of surfactant and then decreases. As the concentration of surfactant increases the amount of foam produced increases, however, during the same time the amount of water decreases. After 80% the foam mass decreases due to the low concentration of water reducing the amount of foam that can be produced. This also shows that commercial surfactant produces the largest mass of foam when is lower water to surfactant ratio, this is true for all sponge types and water types investigated. The effect of salt concentration is apparent for the car sponge whereas for the dishwasher sponge there is no effect on the foamability. For the car sponge the foamability increases as the salt concentration increase this difference in behaviour with respect to water hardness is believed to be due to the how the salts in the hard water solution interact with the surfactants in the dishwashing product allowing for greater penetration into the car sponge.

In Table 1 it is shown that the pore size of the audio sponge is significantly less than that of the car and dishwasher sponges, comparing the foamability of the audio sponge to the dishwasher sponge determines the effect of pore size on formability (Figure 7). It could be anticipated that this difference in pore size would have an impact on the average amount of foam produced. However, results of the average amount of foam produced for the audio sponge and dishwasher sponge presented in Figure 7 demonstrate that there is almost no difference between the foamability of the two sponges, indicating that the difference in pore size has no effect on the amount of foam generated in the range of pore sizes investigated.

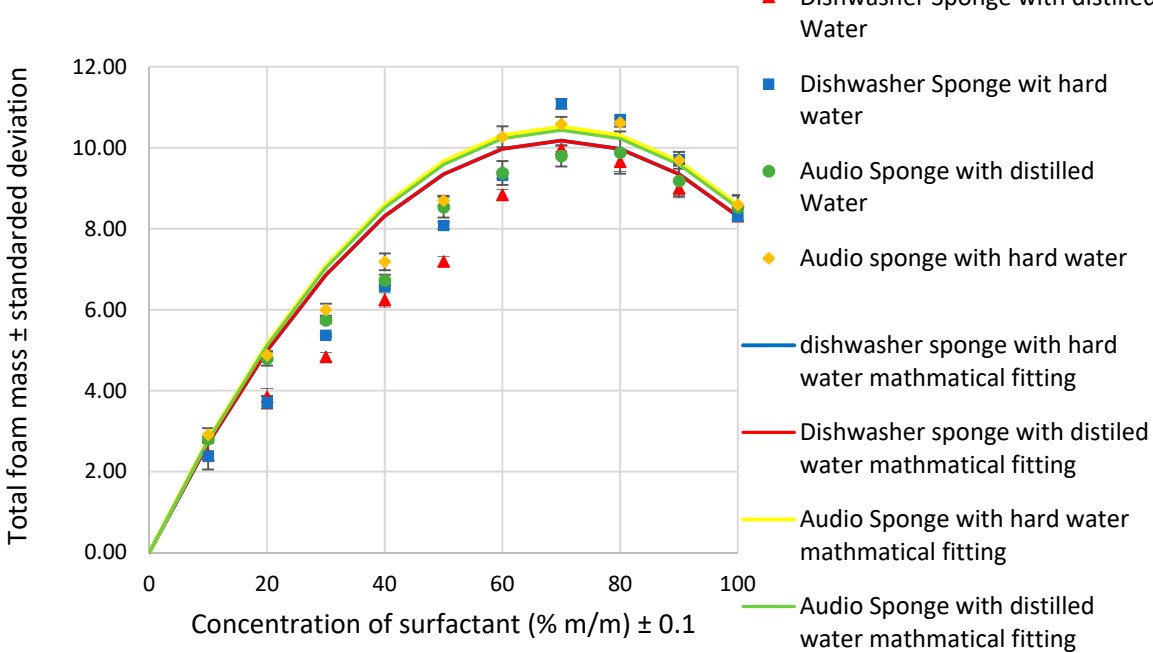

**Figure 7.** Average total foam mass for both the dishwasher sponge and the audio sponge for each concentration of surfactant, fitted according to Equation (2) with $M_{min} = 8.31$ and $c_{max} = 0.7$ for distilled water with a dishwasher sponge, $M_{min} = 8.31$ and $c_{max} = 0.7$ for hard water with a dishwasher sponge, $M_{min} = 8.52$ and $c_{max} = 0.7$ for distilled water with an audio sponge and $M_{min} = 8.59$ and $c_{max} = 0.7$ for hard water with an audio sponge [8].

In conclusion, it was found that for commercial dishwashing surfactant foam mass for all sponges investigated increases until it reaches a maximum at 60–80% of surfactant concentration. The mass of foam then decreases when using pure dishwashing solution. Since at 100% dishwashing solution we have foam formation this indicates that there must be water within the commercial product as well as surfactants and polymers, indicating that "100% commercial surfactant" is not truly pure surfactant solution. Due to the water already in the product the peak at 60–80% that only a slight dilution is required to achieve the maximum amount of foam. indicating that only low levels of liquid core (water) are required to produce the maximum amount of foam. The foamability of the dishwasher sponge and the audio sponge is independent on the hardness of the water used. The foamability of the car sponge increases as the water hardness increases. The difference in behaviour with respect to water hardness is believed to be due to greater penetration into the media for the car sponge. It was also shown that pore size has no effect on foamability.

Commercial dishwashing product is a complex mixture of surfactants and polymers. For better understanding, the system below comparison of foam ability of commercial product and pure SDS suction within porous material was investigated below.

### 3.2. Wetting and Spreading of Commercial Surfactant on Porous Media

To understand a relation between foamability and wetting properties of porous substrates the wetting properties of sponges were investigated. For this purpose, the contact angle and the drop base diameter of each concentration of surfactants were monitored. This work relates to previous work carried out in [26] where the kinetics of wetting where investigated over various substrates. These properties were investigated using KRUSS DSA100 drop-shape analyser. The minimum concentration that could be measured was 60% m/m of commercial surfactant as lower concentration would rapidly absorb before any values could be obtained. The initial contact angle in the case of pure water for

the car sponge was 111°, for the dishwasher sponge the initial contact angle was 116° and the initial contact angle for the audio sponge was 106°. This shows that all the sponges are hydrophobic.

The contact angle and drop base diameter of each of the sponges were compiled into dimensionless form as shown in using the dishwasher sponge as an example, Figure 8. The contact angle values are divided by the maximum contact angle $\theta_M$ and base diameter is divided by the maximum base diameter $L_M$ changing them into dimensionless values $\theta^*$ and $L^*$ respectively. Both $\theta^*$ and $L^*$ are plotted against dimensionless time, $t^*$, which is calculated by dividing the time values by the total time period of the process $t_p$, where the time period is determined by the point the droplet absorbs into the substrate.

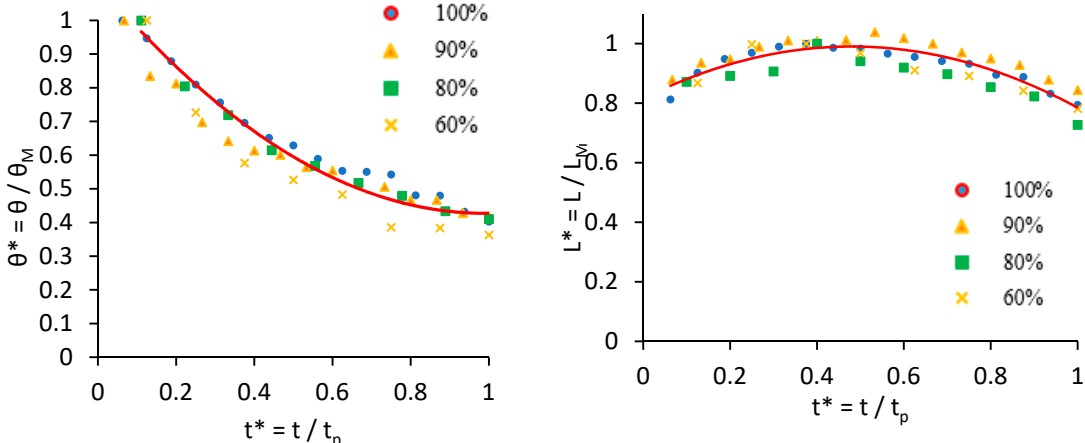

**Figure 8.** Contact angle and droplet base diameter dimensionless profiles for different concentrations of surfactant on dry dishwasher sponge [7].

It was observed that despite the dishwasher sponge and audio sponges having significantly different pore sizes, Table 1, the overall wetting characteristics of the surfactant solutions are identical. It was also found that for all of the sponges there is a minimum contact angle at which any value below this the droplet rapidly absorbs into the surface. This is due to the droplet switching from one wetting state to another from Cassie-Baxter to Wentzel state. This contact angle was found to be around 30° for surfactant solutions on the sponges investigated.

Table 3 shows the initial contact angle of each water type on each of the sponges, it was found that the type of water does not affect the contact angle and hence wetting properties on each of the sponges investigated. For the audio sponge, there are no values as water rapidly absorbed into the media.

**Table 3.** Initial contact angle of droplets of each water type on each sponge type.

| Sponge Type | Initial Contact angle Distilled Water | Initial Contact Angle Hard Water |
|---|---|---|
| dishwasher | 115 ± 8 | 115 ± 8 |
| Car | 111 ± 5 | 111 ± 5 |
| audio | - | - |

Tables 4 and 5 show the initial and final contact angle for 80% dishwashing solution with different water types for each sponge types, Table 4 shows the values when using distilled water and Table 5 shows the values when using 15dH hard water.

**Table 4.** Initial and final contact angle for 80% surfactant solution with distilled water on each sponge type, where the total process time is also shown.

| Sponge Type | Initial Contact Angle | Final Contact Angle | Total Time (s) |
|---|---|---|---|
| dishwasher | 88 ± 1 | 36 ± 1 | 10 |
| Car | 87 ± 5 | 40 ± 8 | 45 |
| audio | 106 ± 1 | 26 ± 1 | 12 |

**Table 5.** Initial and final contact angle for 80% surfactant solution with 15dH water on each sponge type, where the total process time is also shown.

| Sponge Type | Initial Contact Angle | Final Contact Angle | Total Time (s) |
|---|---|---|---|
| dishwasher | 88 ± 1 | 36 ± 1 | 10 |
| Car | 84 ± 6 | 46 ± 6 | 15 |
| audio | 104 ± 2 | 27 ± 4 | 14 |

Table 5 shows that hard water solutions have a greater spread ability on the car sponge which was not observed when using distilled water solutions. This supports what was discussed in [8] that hard water solutions have a greater spread ability and readiness for penetration. Whereas for the audio and dishwasher sponges the hardness water used to create 80% solutions has no effect on the spread ability or penetration into the media.

In conclusion, the method described in this study was proven to be effective characterising the overall wetting and spreading characteristics of the sponges via the generation of universal dimensionless dependences in each case. All the sponges displayed a minimum contact angle where after which the droplet rapidly absolved into the surface.

*3.3. Foamability and Foam Quality of SDS using the Compression Device*

The purpose of the research below is to investigate the foamability of three different sponges (the same as above) with basic SDS surfactant. The foam is created using both distilled and hard water. All these sponges were investigating above as well in [7,8] with commercial surfactant. The main objective is to understand the foamability of a basic surfactant with porous media with the aim to model this process and then in the future apply this to more complex surfactant mixtures. To see the main influencer on foamability when considering foam formation by compression.

Concentration dependency can be fitted using modified Equation (2), which now reads:

$$M(c) = M_{final} + (M_{initial} - M_{final}) \exp (-\gamma(C-C_{initial})) \tag{3}$$

Where $M(c)$ is the foam mass at concentration C, $M_{final}$ is the foam mass at the plateau, $M_{initial}$ is the mass at 0.5 cmc ($C_{initial}$), ɣ is a fitting constant and C is the concentration in terms of multiples of critical micelle concentration. $\alpha$ is obtained using log co-ordinates in Equation (3) shown by Equation (4).

$$\ln\{(M(c) - M_{final})/ (M_{initial}-M_{final})\}=- \gamma(c-c_{initial}) \tag{4}$$

where $\gamma$ is obtained from the gradient of a graph of $\ln\{(M(c) - M_{final})/ (M_{initial}-M_{final})\}$ against $C-C_{initial}$. The values of $\gamma$ are displayed in Table 6.

**Table 6.** The obtained values of γ for each of the sponge types and concentrations of SDS solution using distilled water.

| Sponge Type | γ |
|---|---|
| Dishwasher | 0.6864 |
| Car | 0.2114 |
| Audio | 0.5164 |

For the dishwasher sponge, the effect of increasing surfactant concentration on the mass of foam generated was determined (Figure 9).

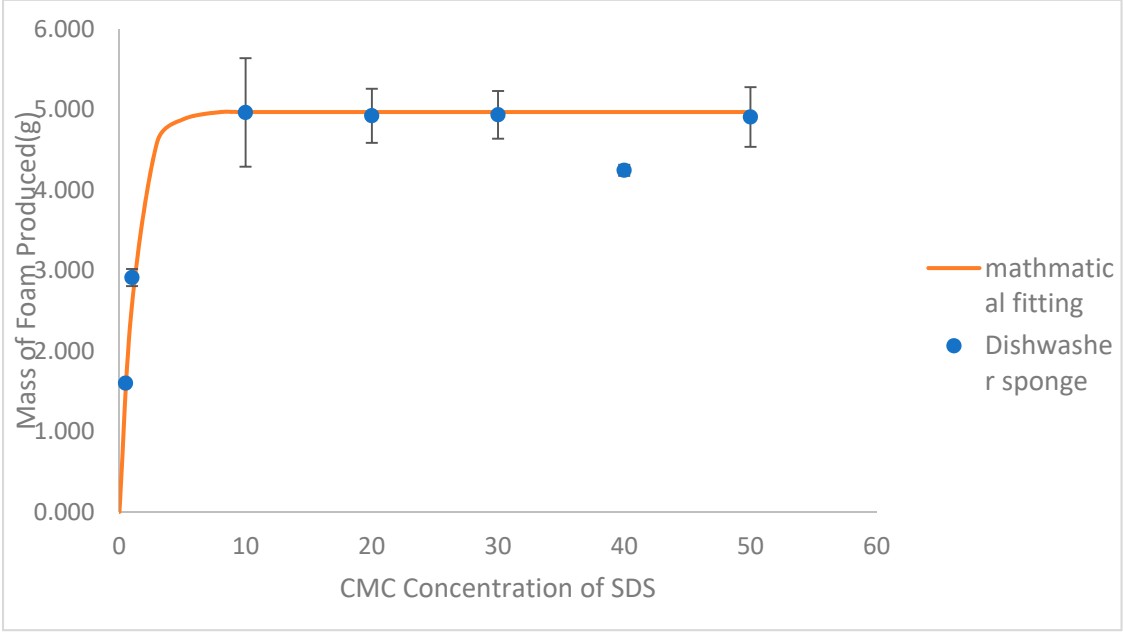

**Figure 9.** The dependency of surfactant concentration of the mass of foam generated by compression where error bars were determined by the standard deviation for the dishwasher sponge. fitting calculated using Equation (3).

Figure 9 shows that as the concentration increases up to the CMC, the mass of foam generated increases and continues to increase until ten times the CMC. The mass of foam peaks at 10 CMC and any further increase in concentration results in no change in the mass generated. The liquid volume fraction is proportional to mass generated as the density of the solution is identical to that of water. Therefore, dividing our mass of foam by the initial mass of solution absorbed into the media (30g) results in the liquid volume fraction, the liquid volume fraction values for each concentration is shown in Table 7.

**Table 7.** Calculated liquid volume fraction for the foam generated for each of the concentrations of SDS solution.

| Concentration (CMC) | Liquid Volume Fraction | ±Error |
|:---:|:---:|:---:|
| 0.5 | 0.053 | 0.001 |
| 1 | 0.097 | 0.004 |
| 10 | 0.166 | 0.022 |
| 20 | 0.164 | 0.011 |
| 30 | 0.165 | 0.010 |
| 40 | 0.142 | 0.002 |
| 50 | 0.164 | 0.012 |

Table 7 shows that the max liquid volume fraction is reached at 10 CMC and then remained constant with further increases of concentration of SDS concentration. This shows that initially increasing the concentration beyond CMC increases the foamability and liquid volume fraction, indicating that some micelles are needed to stabilize the foam structure. The plateau after 10 CMC indicates that any further increase in micelle population after 10 CMC does not increase the stability of the foam leading to constant foamability and liquid volume fraction.

This disagrees with the theory presented by Chiang et al. [17], according to them foam generation increases with concentration up to CMC and above CMC all the additional surfactant will go to forming micelles and not increasing foam generation further. Whereas Figure 9 shows that after CMC, foam generation still increases up to 10 CMC. However, this does agree with past work on this method done by Johnson et al., as the trend shown in Figure 9 is similar to the one presented by Figure 7 in [12], as the mass generated keeps increasing as surfactant concentration increases up to 60% (which is much higher than CMC) and then further increase in concentration results in no increase in foam generation [8].

For each of the sponges, their influence on foam generation across the concentration range was observed. Error bars were removed in order to provide better visualisation (the minimum error is 0.04 g and the maximum error is 0.65 g).

As shown in Figure 10 as the concentration increased from 0.5 CMC to 10 CMC the mass of foam generated increases for all sponge types. For the soundproofing sponge, the mass of foam produced below CMC was much greater than that for both the dishwasher and car sponges. As a consequence, the resulting increase of foam generated from 0.5 CMC to 10 CMC is much greater for the dishwasher and car sponges. As displayed in Figure 9 for the dishwasher sponge the mass of foam produced is not affected by any further increase in surfactant concentration after 10 CMC. This is also true for the audio sponge as shown in Figure 10. Whereas for the car sponge there is a further increase in foam mass produced up to 20 CMC any increase after 20 CMC does not increase the mass of foam produced.

The liquid volume fraction for each concentration for each of the sponge types investigated is displayed in Table 8.

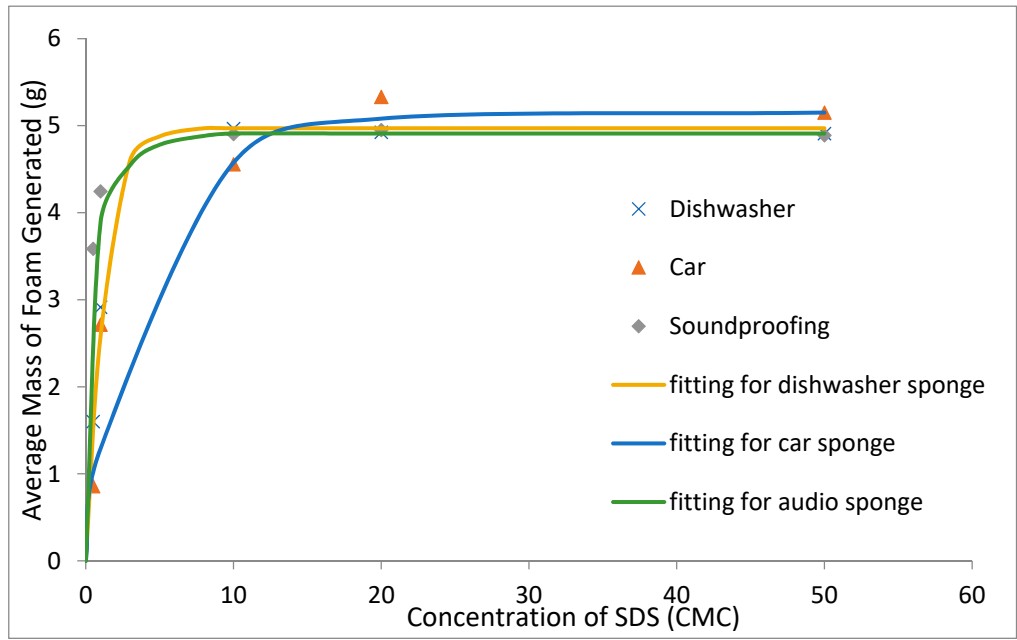

**Figure 10.** The dependency of surfactant concentration on foam generated for the different sponge types investigated using distilled water, where the fitting is calculated using Equation (3).

**Table 8.** Liquid volume fraction for each concentration of each of the sponges investigated using distilled water.

| Sponge Type | Concentration (CMC) | Liquid Volume Fraction | ±Error |
|---|---|---|---|
| | 0.5 | 0.053 | 0.001 |
| | 1 | 0.097 | 0.004 |
| Dishwasher | 10 | 0.166 | 0.022 |
| | 20 | 0.164 | 0.011 |
| | 50 | 0.164 | 0.012 |
| | 0.5 | 0.029 | 0.005 |
| | 1 | 0.090 | 0.002 |
| Car | 10 | 0.152 | 0.013 |
| | 20 | 0.178 | 0.016 |
| | 50 | 0.172 | 0.015 |
| | 0.5 | 0.029 | 0.012 |
| | 1 | 0.142 | 0.006 |
| Audio | 10 | 0.163 | 0.009 |
| | 20 | 0.165 | 0.009 |
| | 50 | 0.163 | 0.022 |

Table 8 shows that the car sponge produces foam with a higher liquid volume fraction and hence a wetter foam than the other two sponges. As the dishwasher and car sponges have similar porosity and pore size it can be concluded that neither of the factors affects foam generation for this method. A factor which will affect the foamability and foam quality is the interconnectivity of the pores which effects the flow rate and air flow through the media in the course of foam generation inside the pores. This property is difficult to measure but can be investigated in the future using CT scanning of the sponge to observe the 3D structure as the solution flows through the media.

## 4. Conclusions

A compression device was developed to investigate the foamability of different sponges.

The pore size and porosity of the car sponge and dishwasher sponges were found to be very similar. The audio sponge's porosity was similar to the car and dishwasher sponge, but the pore size was significantly smaller than the other sponges.

However, the difference in pore sizes does not have a substantial difference in foamability of the dishwasher sponge and audio sponge using commercial surfactants.

Foam mass for all sponges increases until it reaches a maximum at 60-80% of dishwashing surfactant then decreases, which means that only low levels of liquid core (water) are required to produce the maximum amount of foam.

The foamability of the dishwasher sponge and the audio sponge is independent on the hardness of the water used when using a hard water solution with dishwashing surfactant.

The foamability of the car sponge increases as the water hardness increases. The difference in behaviour with respect to water hardness is believed to be due to how the salts in the hard water solution interact with the surfactants allowing for greater penetration into the car sponge.

The method used to determine contact angle and drop base diameter is proven to be effective at characterising the overall wetting and spreading characteristics of the sponges via the generation of universal dimensionless dependences in each case.

All the sponges display a minimum contact angle where after which the droplet rapidly absorbs into the porous media.

Initial contact angle and complete absorption contact angle are dependent on the material and structural behaviour of the media.

Mass generation of SDS foam by compression of a porous material is dependent on surfactant concentration. As the surfactant concentration increases, the mass of foam generated increases even beyond the Critical Micelle Concentration. Increasing the concentration beyond 10 times the critical micelle concentration has no further effect on the amount of foam generated.

The objective of these investigations was to determine what parameters affect the foam generation by compression of a porous material. Expanding on previous work of how foam interacts with a porous media. Further parameters such as water hardness, temperature and pH will be investigated with the SDS solutions used here to understand what effect these have on the amount of foam and structure of the foam produced. Future work on multiple different parameters, such as interconnectivity of pores and material type, can be considered to obtain a full understanding of the main factors of foam production.

**Author Contributions:** P.J.: Methodology, Validation, Formal analysis, Investigation, Data curation, Writing—original draft, Writing—review &editing, Visualization. M.V.: Resources, Writing—review & editing. A.T.: Conceptualization, Validation, Resources, Writing—original draft, Writing—review & editing, Visualization, Supervision, Project administration. V.S.: Conceptualization, Writing—review & editing, Visualization, Supervision. All authors have read and agreed to the published version of the manuscript.

**Funding:** This research was funded by Proctor & Gamble, Brussels; MAP EVAPORATION and PASTA projects, European Space Agency and UKRI

**Acknowledgments:** This research was supported by Loughborough Materials Characterisation Centre, Proctor & Gamble, Brussels; MAP EVAPORATION and PASTA projects, European Space Agency.

**Conflicts of Interest:** The authors declare no conflict of interest.

## Nomenclature

| Symbol | Meaning | Units |
|--------|---------|-------|
| $\Theta$ | Actual contact angle | Degrees |
| $\Theta_M$ | Max. contact angle | Degrees |
| $\Theta^*$ | Dimensionless contact angle | - |
| L | Actual Diameter | mm |
| LM | Maximum diameter | mm |
| L* | Dimensionless diameter. | - |
| t | time | s |
| tp | Total process time | s |
| t* | Dimensionless time | - |
| $M_\infty$ | Maximum foam mass | g |
| c | Concentration of surfactant | - |
| $M_{min}$ | The experimental value for the mass of foam at 100% surfactant | g |
| $c_{max}$ | Surfactant concentration when maximum foam mass is reached | - |
| M(c) | foam mass at concentration c | g |
| $M_{final}$ | foam mass at the plateau | g |
| $M_{initial}$ | mass at 0.5 CMC | g |
| $\gamma$ | Fitting constant | $Cmc^{-1}$ |
| C | concentration in terms of multiples of critical micelle concentration | CMC |

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
