# Peer review of "Foam Formation and Interaction with Porous Media"

_coatings, doi:10.3390/coatings10020143_

Round 1

Reviewer 1 Report

Dear editor, I have received the manuscript  "Foam formation and interaction with porous media" to review.

Here are my comments:

1. The manuscript contains lots of expression of clumsy English, convoluted sentences, and typos. (e.g. lines 12-14, line 16 "60-80m/m of.", line 17 "independent on (of)", line 31 "devise"...). I have problems in understanding the meaning of concentration "10-100 m/m" in line 129

2. There are basic scientific information which are trivial, like the effect of a surfactant (lines 49-51) or the CMC (lines 54-55). A student is supposed to know these information.

3. What is the origin of eq.1? What is the model generating it? In other words,  how was it derived?

4. There is a clear wrong use of significnt digits in table 2. (mind the typo "CaCl12")

5. What is the main clue of the work? How doeas the work advance the science/technology in coating? The Authors should make efforts in clearly show this.

6.  the text is not divided into proper subparagraphs (Introduction, Materials and Methods, Results, Discussion...)

7. Probably the study conducted on three types of sponges only (two of them have the same pore size) is not significant. The effect of pore size (with constant chemical composition) (or the effect of chemical composition at constant pore size) is therefore not unveiled. This would have given some scientific clue.

In the light of the above observation, I think that the work cannot be accepted at the present stage. I suggest, however to consider re-submission after the above items are addressed.

Best regards

Author Response

Thank you for your comments. Please find belong corrections and answers to questions raised.

Reviewer 1

The manuscript contains lots of expression of clumsy English, convoluted sentences, and typos. (e.g. lines 12-14, line 16 "60-80m/m of.", line 17 "independent on (of)", line 31 "devise"...). I have problems in understanding the meaning of concentration "10-100 m/m" in line 129

Corrections to the spelling have been made. Thank you for indication of where some of these mistakes where.

There is basic scientific information which is trivial, like the effect of a surfactant (lines 49-51) or the CMC (lines 54-55). A student is supposed to know this information.

The information mentioned is included to allow an introduction of basic colloid science, allowing people who do not directly deal with surfactants to get a simple introduction to the principles so that they can understand what is discussed in this paper.

What is the origin of eq.1? What is the model generating it? In other words,  how was it derived?

Equation 1 is originally derived in our previous paper ‘Foamability of soft porous media using compression’ (which is reference 8 within the manuscript). Where the experimental results were used to create a mathematical fitting for the data presented.

There is a clear wrong use of significnt digits in table 2. (mind the typo "CaCl12")

Typo corrected and the values rounded up to 2 significant figures.

What is the main clue of the work? How does the work advance the science/technology in the coating? The Authors should make efforts in clearly show this.

The main object of this work was to understand how commercial surfactant and SDS interact with a commercial porous material. Also, what effect if any the different properties of the media has on the foamability. In terms of coating, this manuscript provides a baseline for the interaction of surfactant with an uncoated porous material and the foamability of this media. This will provide a reference to compare with coated porous media. Showing clearly what effect the coating used has on wettability and foamability when used with commercial porous media.

the text is not divided into proper subparagraphs (Introduction, Materials and Methods, Results, Discussion...)

The manuscript has now been rearranged and proper subparagraph titles have been added.

Probably the study conducted on three types of sponges only (two of them have the same pore size) is not significant. The effect of pore size (with constant chemical composition) (or the effect of chemical composition at constant pore size) is therefore not unveiled. This would have given some scientific clue.

The manuscript only considered commercial porous material as the main interest of these experiments was the interaction of surfactant and foamability within a consumer setting. Future work with porous media with well-known chemical and mechanical properties is needed and has been planned. In this experiment the sponges used are stated by the manufacturer to be 100% polyester, meaning that they should have the same chemical composition. This has now been added in the materials and methods section.

Reviewer 2 Report

The paper describes a study about foamability using a sponge compression system. The study may have some interest for specific application. However, the correlation with foam flow application porous media, oil recovery for example, is weak. The authors miss important references and the analysis is only descriptive. Furthermore, the system use for creating the foam is not clear. The foam is created by compression of the sponge with injection of gas. Consequently, it is not recommended to publish this article.

Below some comments:

Line 33 -44 The paper attempts to summarize the utilisation of foam in oil recovery. However the text is confusing and missing the reference paper. Authors should look at publications from majors scientists in the fields like G. Hirasaki and W. Rossen. Please revise the text and cite relevant publications. Line 45 foams in porous media are created by snap off , leave behind and lamella division [kovscek Radke 1996]. The text is wrong and should be corrected. Effect of the surfactant concentration on foam in porous media has been study in the past. For example a complete study in porous media and bulk has been done by Jones et al. 2016 in J Ind. Eng. Chemistry “Effect of surfactant concentration on foam”. Jones et al. found that the optimum concentration is 100 times more than the CMC. Authors (Johnson et al.) found that the optimum concentration is 10 times the CMC in the case of SDS. This difference with Jones et al. should be discussed. Figure 2 , Figure is unclear. The numbers are not clearly readable, please edit the text in the figure. Where is the gas injected? where the foam volume is measured? During the compression, the gas is compressed , decreasing his volume. What is the faction of gas in the sponge? Is it constant? What is the impact on the foam creation? Line 113 114: The calculation of the porosity with a software is dubious. This should be measured weight under vacuum which , knowing the density of the material, can deduce the porosity. Please calculate the porosity in a proper manner. The permeability is important for foam in porous media. Authors should add this information and discuss it in their results. Line 128 129 . the concentration of surfactant goes up to 100% which is very unlikely for SDS. Authors should clarify how to create foam with 100% pure SDS. Figure 5 and others : the concentration of surfactant is in mass but it is not clear in the horizontal label. Please write unit [% mass] or something else. Horizontal axes should go up to 100% (it is not possible to exceed 100% surfactant)

Author Response

Thank you for your comments. Please find belong corrections and answers to questions raised.

Reviewer 2

Line 33 -44 The paper attempts to summarize the utilisation of foam in oil recovery. However the text is confusing and missing the reference paper Authors should look at publications from majors scientists in the fields like G. Hirasaki and W. Rossen. Please revise the text and cite relevant publications.

This section has been reworded to make it easier to follow, there are multiple references within the text that link to enhanced oil recovery.

Line 45 foams in porous media are created by snap off , leave behind and lamella division [kovscek Radke 1996]. The text is wrong and should be corrected.

Line 45 refers to foams in general and the main mechanism of which foams are formed not specifically foam formation in enhanced oil recovery.

Effect of the surfactant concentration on foam in porous media has been studied in the past. For example, a complete study in porous media and bulk has been done by Jones et al. 2016 in J Ind. Eng. Chemistry “Effect of surfactant concentration on foam”. Jones et al. found that the optimum concentration is 100 times more than the CMC. Authors (Johnson et al.) found that the optimum concentration is 10 times the CMC in the case of SDS. This difference with Jones et al. should be discussed.

The main issue with this comparison is that the foam generation methods are different. In our manuscript, the foam is created by compression of a porous material, whereas in the example shown the foam is generated by injecting air into the porous structure. This is similar to experiments we are conducting in a foam column with the sponges investigated. It is was found that the foam production depended heavily on the foam production method used. Finally, the porous media investigated in the example was incompressible hard porous media whereas the porous media in this manuscript was compressible soft porous media. The effect that compressibility and hardness of the media plays is another factor that requires further investigation. Which will become important when trying to consider modelling foam production in porous media.

Figure 2 , Figure is unclear. The numbers are not clearly readable, please edit the text in the figure. Where is the gas injected? where the foam volume is measured? During the compression, the gas is compressed , decreasing his volume. What is the faction of gas in the sponge? Is it constant? What is the impact on the foam creation?

Labels on figure 2 have been edited to make them easier to see. The gas injection is a consequence of the compression of the media by the plates, the foam mass is measured by weighing of foam created by this process. During the compression, the solution saturated sponge is compressed until the minimum volume is reached i.e. the plates come into contact. The foam is created as the gas then flows into the media as the plates are released, leading to the next compression releasing the foam produced. The effects of different volume of air within the media is something being investigated currently which is being conducted by partially squeezing the media. Meaning that not all the porous media will be compressed during the foam production and hence meaningless air can flow into the media.

Line 113 114: The calculation of the porosity with a software is dubious. This should be measured weight under vacuum which , knowing the density of the material, can deduce the porosity. Please calculate the porosity in a proper manner. The permeability is important for foam in porous media. Authors should add this information and discuss it in their results.

The method for measurements of porosity has been used previously in the investigation of porosity of electrospun scaffolds. The software was tested using three electrospun scaffolds and four membranes, where the software for porosity was compared to the experimental values of porosity for these materials. It was shown that there was little discrepancy between the software calculated values and the experimental values of porosity. Reference: Shuai Wang, Glucose diffusivity and spreading experiments with porous scaffolds and membranes for tissue engineering bioreactors, doctoral thesis, Department of chemical engineering, Loughborough University, 2017, page 140-143.

Line 128 129 . the concentration of surfactant goes up to 100% which is very unlikely for SDS. Authors should clarify how to create foam with 100% pure SDS. Figure 5 and others : the concentration of surfactant is in mass but it is not clear in the horizontal label. Please write unit [% mass] or something else. Horizontal axes should go up to 100% (it is not possible to exceed 100% surfactant)

Line 128 and 129 does not relate to SDS the first investigation shown is for commercial dishwashing surfactant and 100% relates to pure dishwashing surfactant solution. The figures have been corrected so that the horizontal axis now finish at 100% and that it now clearly states that its % by mass (m/m).

Reviewer 3 Report

The submitted manuscript entitled ‘Foam formation and interaction with porous media’ deals with the formation and interaction of foams with porous media. The theme is interesting; however, the manuscript is unusual in its structure (no numbered sections) and besides a list of technicalities arose as listed below.

- Please solve each abbreviation at its first occurrence even if it is in the Abstract and if it is well-known.

- There is a typo in the affiliation of the Authors (’Departament’ instead of ’Department’).

- Keywords are missing.

- References in the Abstract are unusual.

- The citation of the References in the main text is not subsequent and starting by [4[.

- The legends and axes labels are too small to read in the figs.

- In fig 2 the labels are missing (except (4)).

- Please always let a space between the value and its dimension except in the case of ‘°C’ and ‘%’.

- The caption of fig 4 is on the next page.

- Please highlight the quality of the fittings (for example R2) in all corresponding cases.

- Tables 6 and 7 are broken.

- Contributions of the Authors are missing.

Author Response

Thank you for your comments. Please find belong corrections and answers to questions raised.

Reviewer 3

Please solve each abbreviation at its first occurrence even if it is in the Abstract and if it is well-known.

The full name for abbreviation has been added.

There is a typo in the affiliation of the Authors (’Departament’ instead of ’Department’).

Thank you for pointing this out this has been corrected

Keywords are missing.

Keywords now added

References in the Abstract are unusual. The citation of the References in the main text is not subsequent and starting by [4].

Abstract references have been removed and the numbering of references has been corrected.

The legends and axes labels are too small to read in the figs.

The sizes of the legends and axes labels have been increased

In fig 2 the labels are missing (except (4)).

Labels in figure 2 where present but difficult to see so has been edited to make them easier to see

Please always let a space between the value and its dimension except in the case of ‘°C’ and ‘%’.

Spaces have been added between units and the numeric value.

The caption of fig 4 is on the next page.

Caption now on the same page as figure 4

Please highlight the quality of the fittings (for example R2) in all corresponding cases.

When considering the experimental error, the r^2 value for the fittings of all the graphs is close to 1 due to the accepted error being greater than the actual error of the plots.

Tables 6 and 7 are broken.

Tables 6 and 7 are no longer broken.

Contributions of the Authors are missing.

Contributions of Authors added.

Round 2

Reviewer 1 Report

Dear Editor I have received the manuscript coatings-680183-revision 2 to review.

I must say that the manuscript, in this revision, looks better.

However:

1. The text still contains lots of oversights and clumsy English expressions. Just as mere examples:

in lines 186-187 "In table 1 it is shown that the pore size of the audio sponge is significantly less than the car and dishwasher sponges" should be like "....than THAT of the car and of the dishwasher sponges"

some repetition: in lines 98-99 "...consists of two parallel plates controlled by pistons which can move to bring the plates together (Figure 2)." is repeated in lines 134-135 "...consists of two parallel plates controlled by pistons which can move to bring the plates together (Figure 2)."

references in paragraph titles are not standard ( [7] at line 211 and [8] at line 142)

in chemical formulas the symbol to indicate the crystallization number of water molecules is a dot (like the symbol for multiplication) and NOT the "full stop" as used by Authors (table 2). The Authors should check. The chemicals used to produce hard water are not mentioned in experimental section ("materials" and methods).

at lines 269 and 270 alpha is discussed but in eq 2 and 3 and table 6 gamma is present

Fig. 6 in X-axis label "surfactants" should be surfactant

Fig. 7 in X-axis label "surfactent" should be surfactant

The Authors should really carefully check the whole text throughout.

2. The Authors in their reply to previous version clarified the origin of equation (1) but not in the manuscript. The origin of eq (1) is still not shown/clarified in the revised version. The Authors are asked to show such information in the manuscript. They also should report briefly the rationale of the derivation of such an equation (i.e. some more details)

3. As regards the scientific aspect, I can notice that in this work the role of surfactant is pivotal. Information on the basic scientific background showing the origin of foam and micelles need to be discussed and commented. This is more true if the authors have chosen to explain even the significance of the term surfactant and CMC. I suggest to include in the bibliography the easy-to-read minireview Colloids and surfaces A: Physicochemical and engineering Aspects (2015), 484, 164-183, taking inspiration from it for comments.

4. The Authors should also be careful when discussing concentrations of surfactant close to 100% since the structure and the dynamics of the fluids can experience big changes. Details on the status of the water and surfactant in such systems are shown in Journal of Molecular Structure (2000) 522, 165-178. The fact that water passes from solvent to solute (and the vice versa is done by the surfactant) implies a turning point making big differences (effect of "lack" of effective water needed to impart to the fluid its original capability of foam forming). Some comments, also related to the "turning point" of the bell-shaped curves shown in the manuscript, are advisable. The Authors can use the suggested paper for inspiration.

In the light of the above comments my opinion is that the manuscript can be accepted after all these issues are addressed

Best regards

Author Response

Reviewer 1

1. The text still contains lots of oversights and clumsy English expressions. Just as mere examples:

in lines 186-187 "In table 1 it is shown that the pore size of the audio sponge is significantly less than the car and dishwasher sponges" should be like "....than THAT of the car and of the dishwasher sponges"

Thank you this has been corrected

some repetition: in lines 98-99 "...consists of two parallel plates controlled by pistons which can move to bring the plates together (Figure 2)." is repeated in lines 134-135 "...consists of two parallel plates controlled by pistons which can move to bring the plates together (Figure 2)."

Thank you, this repeated sentence has been removed.

references in paragraph titles are not standard ( [7] at line 211 and [8] at line 142)

These have been removed and references added into text

in chemical formulas the symbol to indicate the crystallization number of water molecules is a dot (like the symbol for multiplication) and NOT the "full stop" as used by Authors (table 2). The Authors should check. The chemicals used to produce hard water are not mentioned in the experimental section ("materials" and methods).

This has been corrected and this section about hard water has been moved to the materials and methods section.

at lines 269 and 270 alpha is discussed but in eq 2 and 3 and table 6 gamma is present

Thank you it should have been gamma this has been corrected

Fig. 6 in X-axis label "surfactants" should be surfactant

Fig. 7 in X-axis label "surfactent" should be surfactant

Corrections to X-axis have been made

The Authors should really carefully check the whole text throughout.

2. The Authors in their reply to previous version clarified the origin of equation (1) but not in the manuscript. The origin of eq (1) is still not shown/clarified in the revised version. The Authors are asked to show such information in the manuscript. They also should report briefly the rationale of the derivation of such an equation (i.e. some more details)

The reasoning and description of the derivation has now been added to the manuscript.

3. As regards the scientific aspect, I can notice that in this work the role of surfactant is pivotal. Information on the basic scientific background showing the origin of foam and micelles need to be discussed and commented. This is more true if the authors have chosen to explain even the significance of the term surfactant and CMC. I suggest to include in the bibliography the easy-to-read minireview Colloids and surfaces A: Physicochemical and engineering Aspects (2015), 484, 164-183, taking inspiration from it for comments.

Thank you for your reference this has been added and is a very interesting and useful document.

4. The Authors should also be careful when discussing concentrations of surfactant close to 100% since the structure and the dynamics of the fluids can experience big changes. Details on the status of the water and surfactant in such systems are shown in Journal of Molecular Structure (2000) 522, 165-178. The fact that water passes from solvent to solute (and the vice versa is done by the surfactant) implies a turning point making big differences (effect of "lack" of effective water needed to impart to the fluid its original capability of foam forming). Some comments, also related to the "turning point" of the bell-shaped curves shown in the manuscript, are advisable. The Authors can use the suggested paper for inspiration.

This has been addressed as in the manuscript 100% commercial surfactant relates to the commercial dishwashing solution used. At 100% we have mentioned that it is not really 100% pure surfactants but will be a mixture of multiple compounds and since we have foam formation at pure dishwashing product indication that water is already present in the solution already.

In the light of the above comments my opinion is that the manuscript can be accepted after all these issues are addressed

Reviewer 3 Report

Thank you for all the changes and corrections.

Author Response

Thank you for your help.